# Dehydroxylation of Perlite and Vermiculite: Impact on Improving the Knock-Out Properties of Moulding and Core Sand with an Inorganic Binder

**DOI:** 10.3390/ma14112946

**Published:** 2021-05-29

**Authors:** Artur Bobrowski, Karolina Kaczmarska, Maciej Sitarz, Dariusz Drożyński, Magdalena Leśniak, Beata Grabowska, Daniel Nowak

**Affiliations:** 1Faculty of Foundry Engineering, AGH—University of Science and Technology, Reymonta 23, 30059 Krakow, Poland; karolina.kaczmarska@agh.edu.pl (K.K.); dd@agh.edu.pl (D.D.); beata.grabowska@agh.edu.pl (B.G.); 2Faculty of Materials Science and Ceramics, AGH—University of Science and Technology, Mickiewicza 30, 30059 Krakow, Poland; msitarz@agh.edu.pl (M.S.); mlesniak@agh.edu.pl (M.L.); 3Faculty of Mechanical Engineering, Wroclaw University of Science and Technology, 27 Wybrzeże Wyspiańskiego, 50370 Wrocław, Poland; daniel.nowak@pwr.edu.pl

**Keywords:** aluminosilicate, perlite, vermiculite, dehydroxylation, thermal analysis, FTIR, XRD, XRF, SEM, moulding sand, inorganic binder

## Abstract

The article presents the results of research aimed at examining the type of swelling material introduced into moulding or core sand to improve their knock-out properties. Tests on Slovak perlite ore (three grain sizes), Hungarian perlite ore and ground vermiculite (South Africa) were carried out. For this purpose, thermal and structural analyses (FTIR—Fourier Transform Infrared Spectroscopy), a chemical composition test (XRF-X-Ray Fluorescence), phase analysis (XRD—X-Ray Diffraction), and scanning electron microscopy (SEM—Scanning Electron Microscope) as well as final strength tests of moulding sands with the addition of perlite ore and vermiculite were carried out. The results of thermal studies were related to IR (Infrared Spectroscopy) spectra and XRD diffractograms. It has been shown that the water content in the pearlite ore is almost three times lower than in vermiculite, but the process of its removal is different. Moreover, the chemical composition of the perlite ore, in particular the alkali content and its grain size, may influence its structure. The phenomena of expansion (perlite) and peeling (vermiculite) have a positive effect on the reduction of the final sand strength and eliminate technological inconveniences (poor knocking out) that significantly limit the wide use of moulding sands with inorganic binders.

## 1. Introduction

The current state of knowledge indicates that many attempts have been made to eliminate the main technological disadvantage, which is poor knock-out property, that limits the wide use of moulding sands with inorganic binders in a foundry. Taking into consideration the increasing requirements and limitations of environmental protection regulations, it can be predicted that the competitiveness of this type of binder in relation to organic binders will increase. Their wide use is supported by the possibility of obtaining good technological and mechanical properties, the purchase price, much lower than that of organic binders, as well as a small amount of harmful products of thermal decomposition (in the form of gases) generated when pouring molds with a liquid casting alloy and lower costs of waste disposal [1,2]. In addition, the foundries also bear costs related to environmental fees, the amount of which strongly depends on the quantity and quality of generated waste, including gas products. The assessment of the harmfulness of moulding sand does not only concern emissions to the atmosphere. What should also be remembered are the work environment and the frequent exposure of workers to large amounts of harmful substances. To increase the competitiveness and interest of the foundries in inorganic binders, it seems necessary to eliminate or significantly reduce poor knock-out properties. They are the result of the so-called second maximum strength resulting from exposure to a high temperature [3,4]. Many attempts have been made to improve the knock-out properties with the use of various types of materials or by modifying the structure of inorganic binders, e.g., by introducing metal oxides. All these measures were aimed at reducing the final strength of the moulding sand and thus led to an improvement in the knock-out properties [5,6,7,8,9,10,11,12,13].

Knowing the characteristic feature of commonly available materials of mineral origin (perlite ore, vermiculite), which is the increase in volume under the influence of temperature (expansion and exfoliation phenomena), associated with the high speed dehydroxylation reaction [14,15,16,17,18], it was assumed that their introduction into moulding or core sand would reduce the final sand strength and would reduce the technological inconvenience associated with poor knock-out properties. The kinetic energy of the swelling additive located between the grains of the grain matrix will disrupt the continuity of the hardened binder layer and contribute to loosening of the hardened sand and lowering the final strength. Thanks to this, the knock-out will also be improved. Because the minerals belong to the group of inorganic materials, they will not emit harmful gaseous products during heating. In this way, it becomes possible to preserve the basic advantage of sands with inorganic binders—they will remain environmentally friendly.

Based on a literature review, it was found that Greek perlite (Milos Island) is the best studied perlite deposit in Europe [19,20,21]. The paper [21] compares the chemical compositions of Greek, Italian, Hungarian, Chinese, and Turkish perlites and describes the impact of heat treatment processes on changes in IR spectra. However, there is no reference to Slovak perlite and the relationship between the chemical composition and phase changes under the influence of temperature and the results of FTIR structural studies. In the publication [22] on the Slovak perlite ore from the Lehôtka deposit near Brehmi, it was shown that it mainly contains biotite, albite, quartz, and smectite, but the influence of the ore grain size on any changes in the phase composition or changes taking place in the pearlite ore under the influence of temperature (dehydroxylation, chemical composition, phase transitions) have not been described. No detailed studies were found for thermal, structural, and phase analyses for perlite ore from Hungarian deposits beyond the chemical composition listed in [21].

In the works [23,24,25], thermal properties of natural vermiculite were determined. The publication [23,24] concerned the Santa Olalla vermiculite deposit (Huelva, Spain), and tests on samples before and after grinding in a vibratory mill were carried out. In turn, in the publication [25], the authors, referring to the obtained research results, stated that, in their opinion, this process was not sufficiently explained, and therefore a detailed characterization of the structural evolution during the dehydration of high-purity raw vermiculite from the Chinese deposit in Hebei Province using the XRD method in-situ was carried out.

Earlier studies of South African vermiculite from the Palabora deposit showed that it is not pure, as the content of potassium oxide (K_2_O) is higher than 0.35% [26] and contains relatively high levels of iron [27]. It is considered to be one of the most environmentally friendly due to the fact that it is free of asbestos-like fibers and free crystalline silica [28], which is extremely important for the preparation of moulding sand.

In the literature, one can find work on reducing the final strength of moulding sand with an inorganic binder with the use of mineral additives, where the sand was introduced with an additive named Glassex, which is expanded perlite [4,7,10]. A significant gap is the lack of data on the effect of perlite ore with a different grain size and type of deposit on the ability to reduce the final strength of moulding and core sand, and thus improve its knock-out properties. There is also no comparison on the difference between the final strength of moulding sands with an inorganic binder if, at the stage of their preparation, perlite or vermiculite ore is added.

According to the authors, the results contained in the publication supplement the knowledge available so far.

## 2. Materials and Methods

### 2.1. General Characteristics of Materials

Perlite ore from Slovak deposits (SP1–SP3 samples; Perlit AF Sp. Z o.o., Kazimierz Biskupi, Poland) of various grain sizes, Hungarian perlite ore (HP sample; Perlit AF Sp. Z o.o., Kazimierz Biskupi, Poland), and South African vermiculite (V; Vermiculite Poland, Ełk, Poland) were used for the tests. The moulding sands for determination of the tensile strength ($R_{m}^{tk}$) according to the following protocol was carried out:Moulding sand without additives (G)—quartz sand form Szczakowa mine, Sibelco Poland (Bukowno, Poland) (97.28%), inorganic Geopol^®^ binder form Sand Team S.r.o, Holubice, Czech Republic) (2.43%), ester hardener SA72 form Sand Team S.r.o, Holubice, Czech Republic (0.29%);Moulding sand with perlite ore (SP1–SP3 and HP)—quartz sand form Szczakowa mine, Sibelco Poland (Bukowno, Poland) (95.42%), inorganic Geopol^®^ binder form Sand Team S.r.o, Holubice, Czech Republic (2.38%), ester hardener SA72 form Sand Team S.r.o, Holubice, Czech Republic (0.29%), perlite ore (1.91%);Moulding sand with vermiculite (V)—quartz sand form Szczakowa mine, Sibelco Poland (Bukowno, Poland) (96.34%), inorganic Geopol^®^ binder form Sand Team S.r.o, Holubice, Czech Republic (2.41%), ester hardener SA72 form Sand Team S.r.o, Holubice, Czech Republic (0.29%), vermiculite (0.96%).

#### 2.1.1. Perlite Ore

Perlite ore is a transformed effusive rock built of volcanic glass, whose name comes from the French “pearl” (in German “Perlstein”), and is connected with the ball (pearl) form. This form is a result of the stresses caused by the rapid cooling of the glaze. Perlite ore also has a characteristic color and gloss [16,29,30]. Chemically, the perlite is a metastable, amorphous, hydrated potassium-sodium aluminosilicate with a cryptocrystalline structure. In the structure of perlite, apart from chemically bound water, the content of which usually ranges from 2.0 to 5.0% by volume, there may also be quartz inclusions, plagioclase, biotite, and secondary minerals such as montmorillonite or zeolite [30,31]. Deposits of the Slovak perlite are found in tertiary rocks in volcanic areas in the central and eastern part of Slovakia. They are a part of a complex of rocks made of tuffs, rhyolites, and andesites. The resources of these deposits are estimated at over 30 million tons [14]. The tested samples came from the Lehôtka deposit. In turn, Hungarian perlite belongs to the group of Upper Miocene rocks of the rhyolite-rhyodacite type. It was created as a result of undersea volcanic activity on the edge of a subduction zone, with very acidic viscous lava and pyroclastic materials. This ore contains 90–95% of amorphous volcanic glaze, 5.0 to 6.0% of crystalline components, mainly quartz, plagioclase, biotite, and 2.5–3.5% of water [14,32]. Despite comparable prices of these raw materials, perlite from the Hungarian deposits differs from the Slovak perlite ore in terms of properties of the obtained expanded perlite. The Slovak perlite is characterized by a creamier color, higher density, and higher mechanical strength. On the other hand, the Hungarian perlite ore is characterized by a very fine graining and lower mechanical strength [14].

#### 2.1.2. Vermiculite

The name “vermiculite” comes from the Latin word vermiculus and means “worm”. Thus, it refers to the characteristic appearance of the mineral after its rapid heating. Then, the conversion of its internal layers occurs due to the separation of the inter-packet water. The evaporation of the water results in a 15- to 30-fold increase in volume. This process is called exfoliation, i.e., the separation of layers (plates) in grains, and is carried out at a temperature of about 900–1000 °C. Its effect is to increase the volume while reducing the apparent and bulk density [33,34,35,36,37]. In the case of some vermiculites, the beginning of the exfoliation process can be observed at a temperature of 300 °C [36]. As shown in previous studies [34], as a result of rapid heating to 100 °C, vermiculite loses about 50% of the interlayer water, but at this temperature, the layer separation process (exfoliation) does not occur. The vermiculite ((Mg, Fe, Al)_3_(Al, Si)_4_O_10_(OH)_2_·4H_2_O) belongs to the group of clay minerals, which belong to the group of 2:1 type phyllosilicate packets, where the octahedron layer is closed between two tetrahedron layers, vertices facing each other [34,37,38,39,40,41]. Vermiculite is formed as a result of the hydrolysis and weathering of magma packet rocks, hydrothermal action, seepage of groundwater or a combination of these three factors, from dark types of biotite, hydrobiotite, and phlogopite belonging to the mica group [34,35,36,42]. Vermiculite is very rarely present as a pure component, and most often it forms mosaic-like mixed-pack structures. Depending on the deposit and its impurities, its color ranges from yellow-golden, brown to olive [34,43,44,45].

### 2.2. Characterization of Methods

The following devices and research methodology were used in the research:The perlite ore samples did not require preparation. In turn, vermiculite was subjected to the process of grinding (comminuting) in a ball mill in order to obtain a fine-grained fraction, because only the material prepared in this way is suitable for use in moulding sands.Sieve analysis on a standard set with a mesh size from 0.056 to 1.6 mm (average of two measurements), in accordance with the Polish Standard (PN–85/H–11001) was carried out.Simultaneous TG/DTG/DTA (Thermal Gravimetry/Differential Thermal Gravimetry/Differential Thermal Analysis) thermal examinations of the tested materials were performed with use of the Thermal Analyzer produced by Jota (Kraków, Poland). The temperature range of test was 20–1000 °C, and the heating rate was 10 °C/min in an air atmosphere in alumina pans.Structural studies were carried out by means of infrared spectroscopy (FTIR) by a transmission technique consisting of preparing pellets in potassium bromide (KBr), using an Digilab Excalibur FTS 3000 (Bio Rad, Hercules, CA, USA) spectrometer with a standard DTGS detector. The spectra were recorded in the Resolution Pro (Varian/Agilent Technologies, Santa Clara, CA, USA) program in the specific infrared range (4000–400 cm^−1^), with a resolution of 4 cm^−1^. In order to obtain the appropriate quality of the spectra, the samples were ground in the form of fine powder in agate mortars and then mixed with KBr in the ratio 1:100. Test material samples were obtained as a result of rapid heating in a silite furnace at a temperature in the range of 100–1000 °C. The residence time of the sample in the oven was 10 min in each case.The chemical composition of the investigated materials was determined by the X-ray Fluorescence Spectrometry (XRF) method, using a WD-XRF Axios Max spectrometer with Rh 4 kW PANalytical lamp (Malvern Panalytical, Malvern, UK).To analyze the phase composition of the materials, a Philips/Panalytical X’Pert Pro MD powder diffractometer (Malvern Panalytical, Malvern, UK) using Cu Kα1 radiation was used. Standard Bragg-Brentano geometry with a θ–2θ setup was applied (0.008° step size and 5–90° 2θ range). Highscore Plus 3.0 software with a database (Version: PDF-4+2021 powder diffraction database, 2021, International Centre for Diffraction Data, Newtown Square, Pennsylvania, USA) was used to determine the positions of the observed peaks and assign them to the appropriate phases.Moulding sands with an inorganic geopolymer binder were prepared in a roller mixer. First, sand without additives (without perlite ore and vermiculite) was prepared, which was the reference point. Then, the moulding sands with additives were prepared. The mixing time was 5 min in each case. The loose components, i.e., the grain matrix (quartz sand) and the perlite ore/ground vermiculite (mixing time 1 min) were mixed first, then the SA72 hardener (Sand Team S.r.o, Holubice, Czech Republic) (mixing time 1 min) was added, and the (Geopol binder Sand Team S.r.o, Holubice, Czech Republic) (mixing time 3 min) was added in the last stage.Standard fittings for determining the tensile strength were made of the prepared sands. A portion of the moulding sand was placed in a special matrix, coupled with a device for a vibratory concentration of samples (LUZ-2e apparatus manufactured by Multiserw Morek, Brzeźnica, Poland). The hardening process in the open air, under laboratory conditions (temperature 20–21 °C, relative humidity 30–35%) was carried out. After hardening the moulding sand (within 24 h), the samples were heated together in the furnace to a temperature in the range of 100–1000 °C, with a heating rate of 10 °C/min (the same as in the case of the thermal analysis). After the sand samples and the furnace had cooled down to the ambient temperature, the tensile strength ($R_{m}^{tk}$) was determined using the LRu-2 moulding sand strength tester (Multiserw-Morek, Brzeźnica, Poland). Moulding sand without additives (as a reference point) (G), sands with the addition of the Slovak perlite ore of various grain sizes (SP1–SP3), the Hungary perlite ore (HP), and crushed vermiculite (ground) (V) were also made.Microscopic images of moulding sands with additives were made with the use of a HITACHI TM-3000. A Hitachi TM3000 scanning electron microscope (SEM) (Hitachi High-Tech Co., LTD, Tokyo, Japan) was used in this study to investigate the surface morphology of specimens, with a 30 nm resolution, a charge reduction mode of 5 or 15 kV voltage, and 15× to 30,000× magnification capabilities. For high-resolution imaging, the samples were carbon sprayed using a Quorum Techologies Q150T high vacuum sputter (Lewes, UK).

## 3. Results and Discussion

### 3.1. Sieve Analysis

In the first stage of the research, sieve analysis of materials was carried out. Table 1 and Table 2 show their grain size composition and a comparison of the basic grain size parameters.

### 3.2. Thermal Analysis

Figure 1, Figure 2 and Figure 3 show the results of the thermal analysis for the Slovak perlite ore of various grain size (SP1–SP3), and Figure 4 shows the results of the determination for the Hungarian perlite ore (HP). In turn, the results of the analysis for ground vermiculite (V) are shown in Figure 5.

The endothermic effect with a maximum at a temperature of approx. 100 °C indicates the ongoing dehydration process (weight loss up to 190 °C). Slovak perlite ore then loses a small amount of adsorbed water (SP1—0.19%, SP2—0.14%, SP3—0.14%), while Hungarian perlite loses only 0.05%. The temperature of maximum weight loss for SP1–SP3 was approx. 85 °C and for perlite it was 75 °C. As a result of further heating from 190 °C to 1000 °C, about 3% weight loss was noted for the Slovak pearlite ore samples. Slightly greater weight loss was recorded for the Hungarian pearlite ore sample, amounting to 3.29% (Figure 4). The dehydroxylation process takes place over a wide temperature range, which proves the presence of several types of water (free, structural) in the perlite ore structure. The gradual heating is therefore not conducive to the rapid dehydroxylation reaction of the perlite ore associated with a significant increase in volume, as part of the “energy charge” accumulated in the material structure is released in stages. No significant relationship was found between the grain size of Slovak perlite ore (with a different main fraction) and the amount of water accumulated in its structure.

On the TG-DTG curves obtained for the ground vermiculite sample (Figure 5), it is possible to indicate four stages of weight loss. On the DTA curve there are four endothermic and two exothermic effects. At the temperature of 195 °C, there is a first loss of weight about 3.83% (maximum weight loss rate at 152 °C) and it should be related to the reaction of removing the water adsorbed on the surface of the material (endothermic effect at 149 °C). A second weight loss was recorded at a temperature range 195–430 °C, which is related to the reaction of removing the inter-packet water and water associated with the cation exchange (endothermic effect with maximum at 255 °C). The beginning of the dehydroxylation process, characterized by a slow rate associated with the release of OH groups bound in the vermiculite structure, occurs at the third and fourth stage of weight loss at temperatures above 500 °C. The third weight loss (1.87%) occurs in the range of 500–800 °C with a maximum weight loss rate at 742 °C. The fourth, in the range of approx. 850–1000 °C with a maximum weight loss rate at 900 °C, is associated with a weight loss of approx. 2.04%. The appearance of exothermic thermal effects around 940 °C and 980 °C indicates a change in the initial structure of the vermiculite and the crystallization of a new phase. Between them, an endothermic heat effect (approx. 959 °C) was recorded, which may indicate dehydroxylation of the residual hydroxyl groups. These changes inspired the authors to conduct infrared (FTIR) structural research and phase composition analysis (XRD).

### 3.3. X-ray Fluorescence Analysis (XRF)

In order to determine the chemical composition of the starting materials and to properly interpret the results of X-ray diffraction (XRD), tests were performed using X-ray fluorescence spectroscopy (XRF). Table 3 and Table 4 show the elemental and oxide compositions, respectively.

Based on the analysis of the chemical composition, it was found that the size of the Slovak perlite ore fraction influences the content of some oxides. As indicated in the work [46], the K_2_O/Na_2_O ratio is particularly important from the point of view of the expansion susceptibility of perlite ore. As shown in Table 4, the K_2_O/Na_2_O ratio for the smallest fraction (SP1) is 4.17, for the average 4.36, and for the coarsest 1.20. On the other hand, for the Hungarian perlite ore, which, according to the grain composition analysis, has the main fraction collected on the same sieves as the medium-grained perlite (SP2), the K_2_O/Na_2_O ratio is 2.83. It should be noted that the content of K_2_O, CaO, Fe_2_O_3_, and TiO_2_ oxides decreases with the increase in the grain size of Slovak pearlite, while the share of SiO_2_, Na_2_O, and MgO increases. Hungarian perlite has the lowest MgO and TiO_2_ contents. Turkish perlite [47] has a similar content of SiO_2_ (approx. 71%) and Al_2_O_3_ (13%), but it contains fewer Fe_2_O_3_ inclusions (approx. 1.6%). Compared to Macedonian perlite [48], which is considered rich in K_2_O and Na_2_O oxides (4.21% and 3.56%, respectively), Slovak and Hungarian perlite in particular has an even higher K_2_O content, even above 8%, in the case of the medium fraction (SP2) and above 7%—the smallest fraction (SP1).

Ground vermiculite (V) is characterized by a high content of SiO_2_ and Fe_2_O_3_ (34.12% and 29.88%, respectively). It also has a high K_2_O content (approx. 8.5%), which is consistent with the literature [37], but the MgO content is still higher and amounts to 14.27%. Compared to the Iranian vermiculite from the province of Gilan [49], which has a similar MgO content, a significant difference should be noted in the content of Al_2_O_3_—Iranian 15.70% in relation to 6.55 and Fe_2_O_3_—12.90% to 29.88%. In the Iranian vermiculite, the content of the remaining components is: K_2_O = 0.97%, CaO = 5.70%, Na_2_O = 0.16%. The SiO_2_ content is at a similar level.

### 3.4. Structural Analysis (FTIR)

Figure 6, Figure 7, Figure 8, Figure 9 and Figure 10 show the IR spectra of the tested materials exposed to the temperature in the range of 100–1000 °C (prepared in an oven).

Based on the IR spectra of Slovak perlite samples, rapidly heated in a furnace at a fixed temperature, it can be concluded that the water removal process runs over a wide temperature range. Initially, there is a visible weakening of the intensity of the band in the wavenumber range 3650–3400 cm^−1^ and the band occurring around 1640 cm^−1^. The dehydroxylation process associated with the removal of OH groups takes place up to the temperature of about 800 °C. For the sample of Slovak perlite with the smallest grain size (SP1), the primary structure of the material decomposed at 1000 °C (Figure 11). The remaining samples, both Slovak and Hungarian perlite ores, retain their structure, but a change in the half-width of the bands can be indicated as the temperature increases. The analysis of the spectra of vermiculite (V) shows that at 500 °C, water is removed from its structure, which is manifested in the disappearance of the characteristic bands within the wavenumbers 3440 cm^−1^ (stretching vibrations νOH) and 1643 cm^−1^ (deformation vibrations δOH). Additionally, at a temperature of 900 °C, changes in its structure begin, initially manifested by the appearance of two bands: at 878 cm^−1^ and 839 cm^−1^, along with an increase in their intensity in the spectra of the sample exposed to the temperature of 1000 °C and a clear shift of the maximum intensity of the main band from 1002 to 1032 cm^−1^, characteristic of Si–O asymmetric stretching vibrations [50,51]. According to literature data [52] at 900 °C the illite decomposes and enstatite appears, and when heated to 1000 °C, enstatite decomposes and forsterite appears. Such a course is indicated by the appearance of an intense band at 878 cm^−1^, characteristic of forsterite [53].

### 3.5. Analysis of the Change in the Phase Composition (XRD) under the Influence of Temperature

Greek perlite (Milos, Tsigrado, and Trachilas fields) was investigated in the publication [20]. XRD analysis of the starting perlite revealed the presence of glassy phase, quartz, feldspar K, plagioclase, and mica. Small amounts of ilmenite, zirconium, and spinel were also identified, but only in the Tsigrado pearlite sample, which were removed in the expansion process. In turn, the publication [19] showed that Greek perlite consists mainly of silicate glass (amorphous) and additives in the form of biotite, quartz, and feldspar. In the case of Macedonian perlite [48], XRD analysis showed the presence of a large amount of aluminosilicate amorphous and small amounts of crystalline, represented mainly by feldspar, quartz, and cristobalite. Feldspar is represented as K-plagioclase, Na-feldspar, and microcline. The less pronounced presence of SiO_2_ polymorphs represented as α–quartz and cristobalite was also indicated. Compared to the original perlite, the expanded one showed a marked increase in the amount of cristobalite present.

The phase composition tests were carried out at the characteristic points obtained in the thermal analysis and in the FTIR structural tests. Figure 11, Figure 12, Figure 13 and Figure 14 show the diffraction patterns for the tested samples of Slovak perlite (SP1–SP3) and Hungarian (HP) ore. Figure 15 shows the diffractograms for ground vermiculite (V).

The finest fraction of Slovak perlite (SP1) in the initial state consists mainly of the amorphous phase—this is indicated by the presence of the so-called amorphous halo in the range of 2θ = 18–30°, with visible inclusions of the crystalline phase represented by cristobalite (C), albite (A), illite (I), biotite (B), and calcium and aluminum silicates (S)—Figure 11. At the temperature above 300 °C, no changes in the phase composition were found, while above the temperature of 900 °C, reflections from the crystalline phases disappear, and the sample consists only of the amorphous phase, which confirms the conclusions of the spectroscopic studies (Figure 6) regarding the disintegration of the structure. The mean fraction (SP2) and the coarsest fraction (SP3) have a phase composition similar to that of the SP1 sample. In addition to the amorphous halo, reflections related to the presence of albite (A), cristobalite (C), and illite (I) were identified, but no biotite (B) was found, which was probably screened entirely to the finest fraction (SP1). At a temperature above 300 °C, the phase composition of the SP2 and SP3 samples also did not change, and at a temperature above 900 °C, two phases were identified in the SP2 perlite: albite (A) and cristobalite (C), while in the SP3 sample there was a clear increase in the amount of the amorphous phase and the formation of a new phase—spinel (Sp), most likely as a result of illite (I) decomposition [54].

The Hungarian perlite (HP) is the most homogeneous in terms of phase composition. The presence of the amorphous phase (amorphous halo), cristobalite (C), and albite (A) was recorded. During heating, at a temperature above 300 °C, a decrease in the intensity of the amorphous halo associated with the amorphous phase is visible, and at the same time, an increase in the intensity of reflections from cristobalite (C). Above 900 °C, the presence of cristobalite (C) and albite (A) was found.

Three main phases were distinguished in the sample of the starting vermiculite after grinding: vermiculite, hydrobiotite, and biotite. As a result of heating the sample to the temperature of 700 °C, the phase composition does not change. Only above 800 °C do the reflections from vermiculite disappear, which proves the decomposition of layered minerals, and on the diffractograms obtained for samples annealed at 800 and 900 °C, two phases were distinguished—biotite and enstatite. Due to the clear changes in the structure noticed in the MIR (Mid-infrared) spectrum at 1000 °C (Figure 10), a phase analysis was performed for the sample annealed at this temperature. The results of the phase analysis (Figure 15) are consistent with the conclusions presented on the basis of spectroscopic studies. The clear shift in the position of the band associated with the Si–O stretching vibration can be related to the appearance of iron-aluminum spinel, in which aluminum is present in coordination 6 with the formation of forsterite.

### 3.6. Investigation of the Final Strength ($R_{m}^{tk}$) of Moulding Sands with Mineral Additives

The knock-out tests determine the susceptibility of the moulding sand to fragmentation under the influence of mechanical impact and removal from the mold (moulding sand) or casting (core sand) after the casting has cooled down to the knock-out temperature. Assessment of the knock-out properties of the moulding sand was carried out by determining the final strength, which allows capturing the first and second maximum strength (the first at a temperature of 200–300 °C, the second in the range of 800–900 °C). The research was aimed at showing the effect of the type of material on the obtained final strength depending on the annealing temperature. The figure shows the results of the determination for moulding sand without additives (G), sand with the addition of the Slovak perlite ore (SP1–SP3), the Hungary perlite ore (HP), and the ground vermiculite (V).

Figure 16 shows that the moulding sand with a geopolymeric binder without additives (G) shows an increase in strength at 200 °C (first hardening).

The second maximum strength occurs at the temperature of 800 °C and is 4.95 MPa. The introduction of perlite ore into the moulding sand, regardless of the fraction size, has a positive effect on the reduction of the final strength, especially at higher temperature. The final tensile strength ($R_{m}^{tk}$*)* with the addition of SP2 and SP3 Slovak perlite ore at the temperature of 800 °C decreases to about 0.4 MPa. Even better results are obtained by the moulding sand with the addition of the finest fraction (SP1), for which the final tensile strength value $R_{m}^{tk}$ = 0.29 MPa was obtained. In the case of the use of Hungarian perlite ore (HP), a visible decrease in final strength was also noted compared to the moulding sand without additives (G), but the strength at 800 °C is twice as high as in the case of moulding sand with the addition of Slovak pearlite ore of similar grain size (SP2). A better effect is obtained at a higher temperature, i.e., 900 °C, which, according to the authors, should be associated with a generally higher content of water accumulated in the structure of Hungarian perlite (3.29% according to thermal analysis) released at a higher temperature. The desired effect of lowering the final strength obtained by introducing vermiculite into the moulding sand is achieved at a much lower temperature than in the case of perlite ore. The moulding sand with vermiculite (1.0 mass parts.), already at the temperature of 500 °C, has a final tensile strength less than 0.01 MPa. This additive effect is due to the lower dehydroxylation temperature.

An important aspect from the technological point of view is the selection of the fraction of additives depending on the size of the matrix of the moulding sand. Proper adjustment of the size of the additive allows for better filling of voids between grains (pores). The greater degree of fragmentation means that they are able to be distributed throughout the entire volume of the moulding sand, which, when heated, gives the sand gives better results in terms of the number of microcracks and translates into a lower value of final strength.

The chemical composition of individual fractions may also influence the swelling process. The finer fraction of Slovak perlite is characterized by a higher K_2_O/Na_2_O ratio. Perhaps it is an additional factor determining the reduction of the final strength of the moulding sand with SP1 ore.

Additionally, positive results are obtained with a lower share of vermiculite in the moulding sand (1.0 mass parts). As shown in previous studies [55], vermiculite added to the moulding sand in the initial form (nor ground) causes its destruction to an excessive degree. 

### 3.7. SEM Microscopic Imaging

The microscopic tests were carried out to illustrate the mechanism of action of swelling additives in moulding sands with an inorganic binder, i.e., to capture microcracks resulting from increasing the volume of additives located in the inter-grain pores of the quartz matrix. Figure 17 and Figure 18 show the images made with the use of an electron microscope (SEM) for selected samples of moulding sands subjected to temperature influence: with the addition of Slovak perlite (SP1) ore and ground vermiculite (V).

The SEM photos presented above show that the additives introduced during the preparation of the moulding sand (perlite ore, vermiculite) are located between the grains of the moulding sand matrix. Under the influence of the high temperature of the casting alloy, the mineral additives swell, and the energy released in this process is transferred to the adjacent matrix grains, covered with a layer of binder, causing microcracks. The broken binder continuity reduces the final strength of the moulding sand, and thus contributes to the elimination of the technological inconvenience of poor knock-out. As a result, the efficiency of castings production is improved and energy and labor costs are reduced, without harming the natural environment.

## 4. Conclusions

Based on the research, the following conclusions were drawn:on the basis of the conducted thermal analysis, no significant correlation was found between the grading of the Slovak perlite ore (SP1–SP3) and the amount of water accumulated in its structure;the chemical composition, in particular the alkali ratio K_2_O/Na_2_O, and the size of the fraction may have a significant impact on the expandability of the pearlite ore. In the case of the finest fraction at a temperature above 900 °C, it is possible to create an amorphous phase in the entire sample volume;in the finest fraction of Slovak perlite (SP1) at the temperature above 300 °C, no changes in the phase composition were found, while above the temperature of 900 °C, reflections from the crystalline phases disappear, and the sample consists only of the amorphous phase.at temperatures above 900 °C, two phases were identified in the SP2 perlite: albite and cristobalite, while in the SP3 sample, the formation of a new phase—spinel was found, most likely as a result of illite decomposition.Hungarian perlite is characterized by greater stability in terms of phase composition during heating in the tested temperature range. Above 900 °C, the presence of cristobalite and albite was found;vermiculite has a much higher water content than perlite ore—regardless of grain size and origin. The formation of the spatial structure characteristic of vermiculite, where the layers separate and there is a significant increase in volume, is observed at a temperature of approx. 500 °C;it was shown that the structure of ground vermiculite does not change until a temperature of about 700 °C. Only above 800 °C do the reflections from vermiculite disappear, which proves the decomposition of layered minerals, and the diffractograms obtained for samples annealed at 800 and 900 °C distinguish two phases—biotite and enstatite;in the IR spectrum obtained for a sample of vermiculite annealed at the temperature of 1000 °C, a clear shift in the band position related to the Si–O stretching vibration was shown, which was interpreted as the appearance of an iron-aluminum spinel in which aluminum is in 6 coordination with the formation of forsterite;the processes of expanding perlite ore and exfoliating vermiculite have a positive effect on reducing the final strength of moulding and core sand with inorganic binders used in the foundry, eliminating technological inconveniences—poor susceptibility to knocking out. However, in the considered variants, the best effects of lowering the final strength of moulding sands with an inorganic geopolymer binder were obtained in the case of introducing the SP1 Slovak perlite ore in the finest fraction, which correlates well with the conclusion presented above.

## Figures and Tables

**Figure 1 materials-14-02946-f001:**
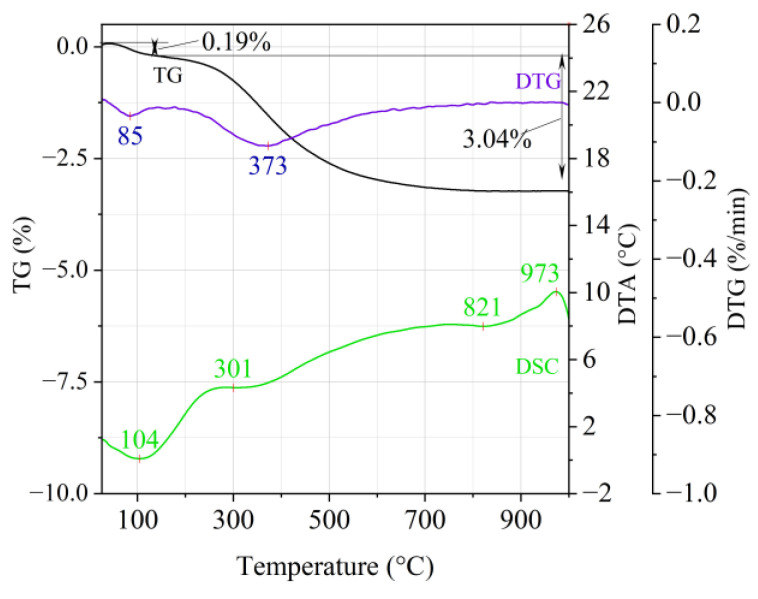
Thermal analysis curves of Slovak perlite ore (SP1).

**Figure 2 materials-14-02946-f002:**
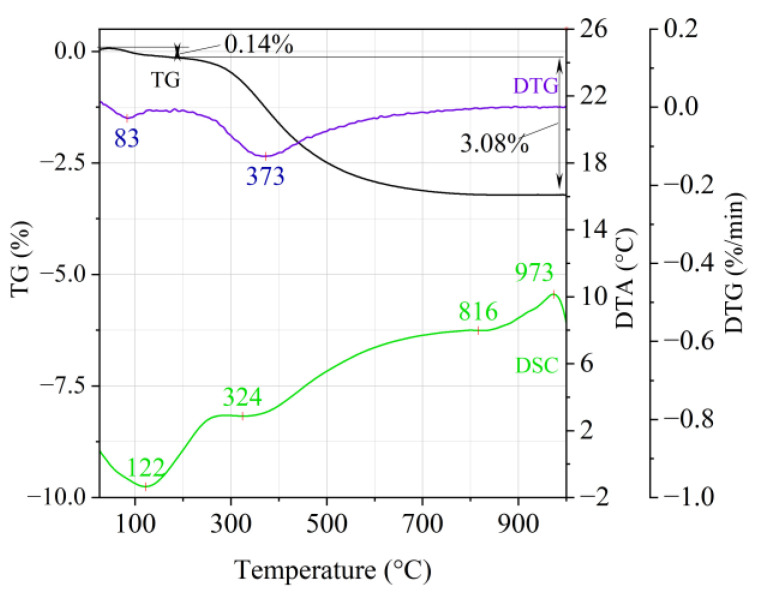
Thermal analysis curves of Slovak perlite ore (SP2).

**Figure 3 materials-14-02946-f003:**
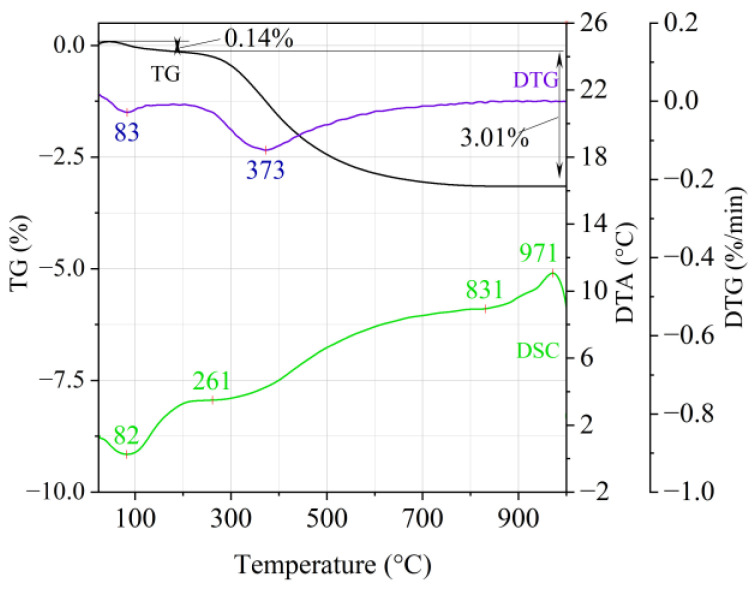
Thermal analysis curves of Slovak perlite ore (SP3).

**Figure 4 materials-14-02946-f004:**
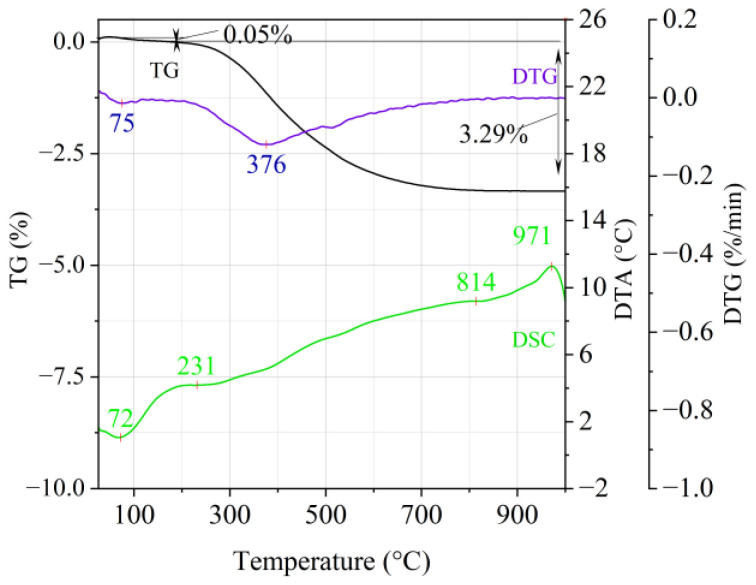
Thermal analysis curves of Hungarian perlite ore (HP).

**Figure 5 materials-14-02946-f005:**
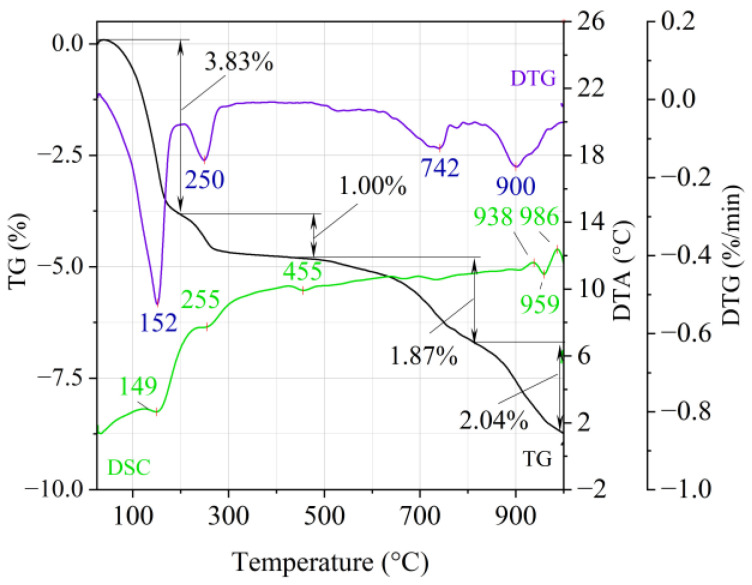
Thermal analysis curves of ground vermiculite (V).

**Figure 6 materials-14-02946-f006:**
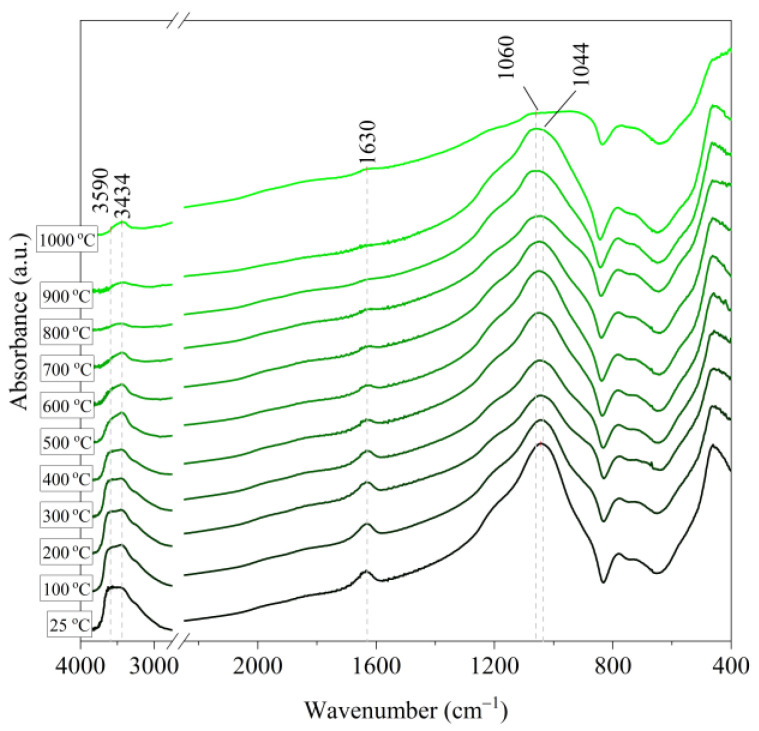
IR spectra of Slovak perlite ore (SP1) after annealing at a temperature in the range of 100–1000 °C.

**Figure 7 materials-14-02946-f007:**
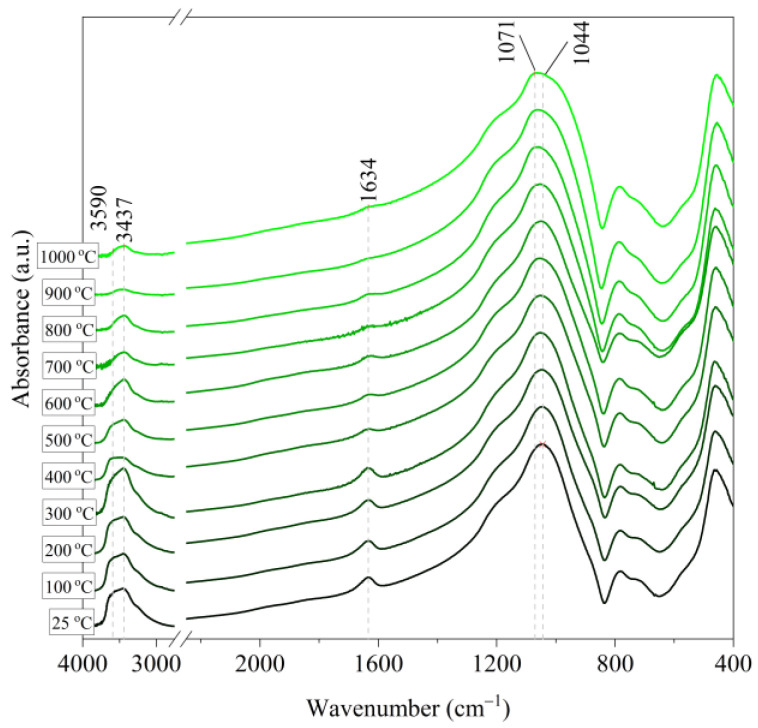
IR spectra of Slovak perlite ore (SP2) after annealing at a temperature in the range of 100–1000 °C.

**Figure 8 materials-14-02946-f008:**
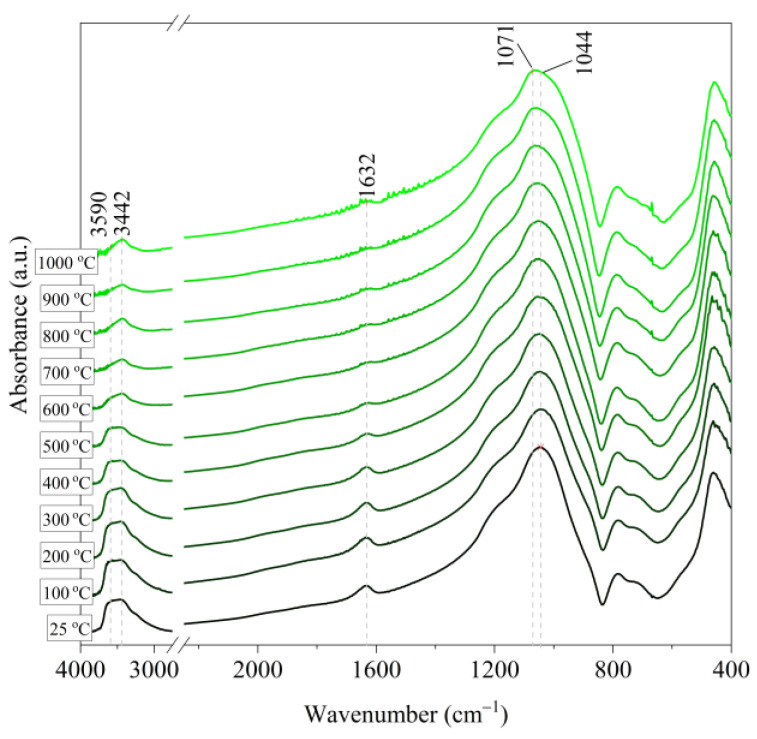
IR spectra of Slovak perlite ore (SP3) after annealing at a temperature in the range of 100–1000 °C.

**Figure 9 materials-14-02946-f009:**
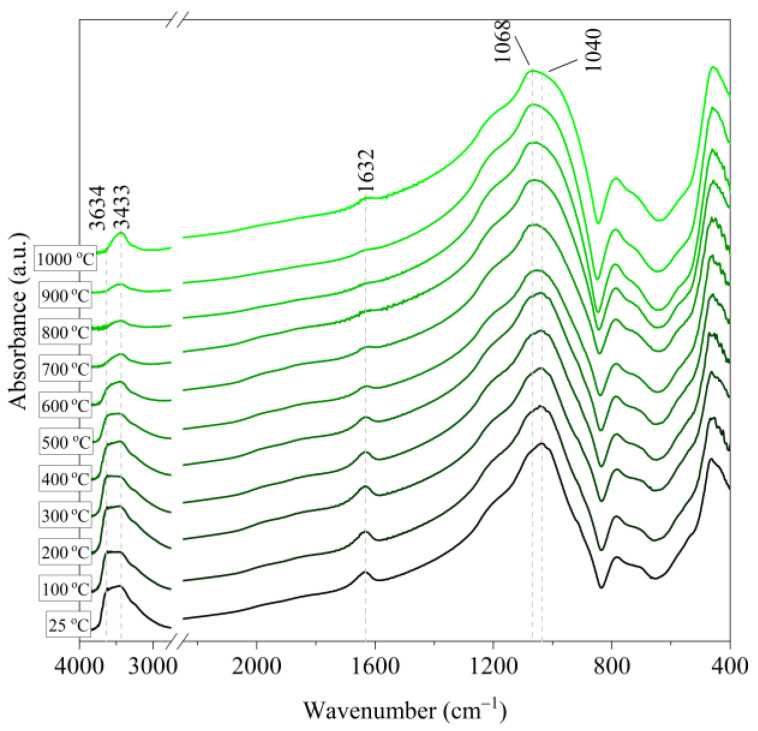
IR spectra of Hungarian perlite ore (HP) after annealing at a temperature in the range of 100–1000 °C.

**Figure 10 materials-14-02946-f010:**
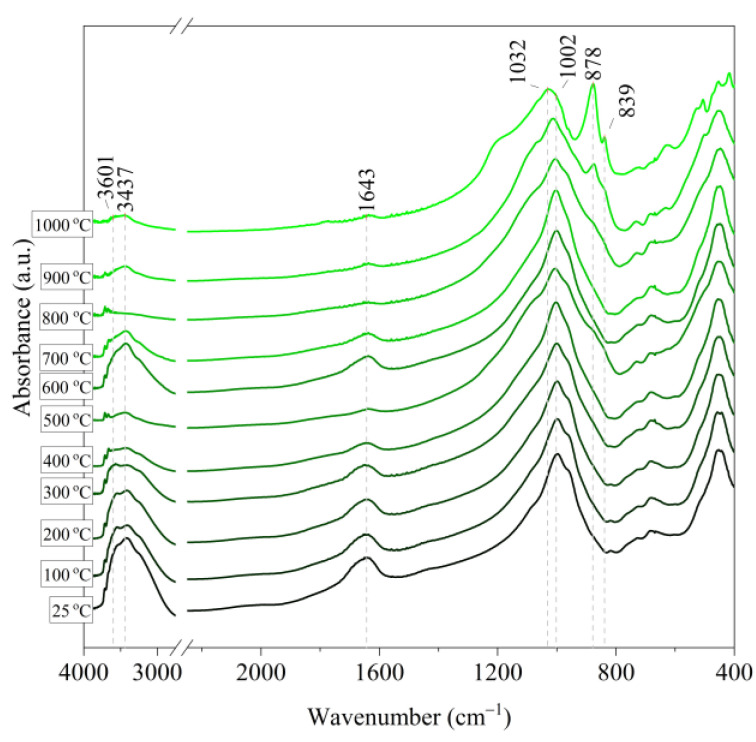
IR spectra of vermiculite (V) after annealing at a temperature in the range of 100–1000 °C.

**Figure 11 materials-14-02946-f011:**
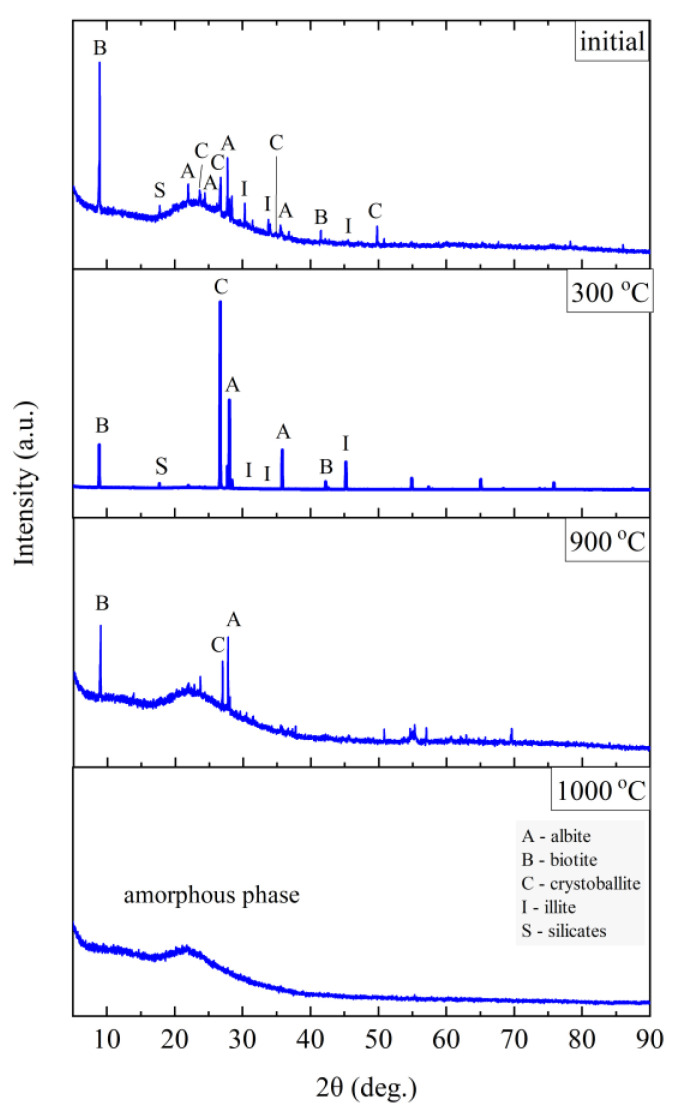
XRD diffractograms for Slovak perlite ore SP1.

**Figure 12 materials-14-02946-f012:**
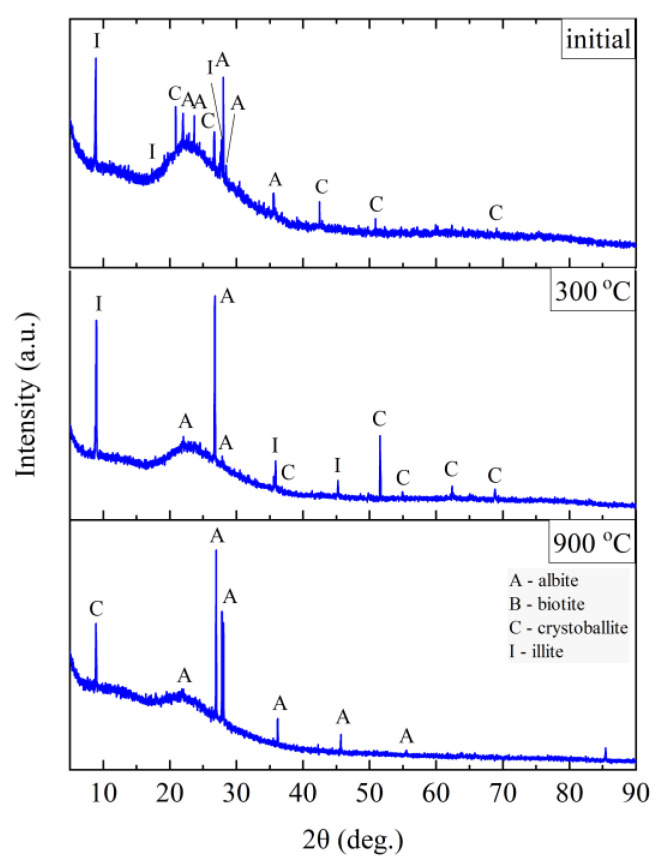
XRD diffractograms for Slovak perlite ore SP2.

**Figure 13 materials-14-02946-f013:**
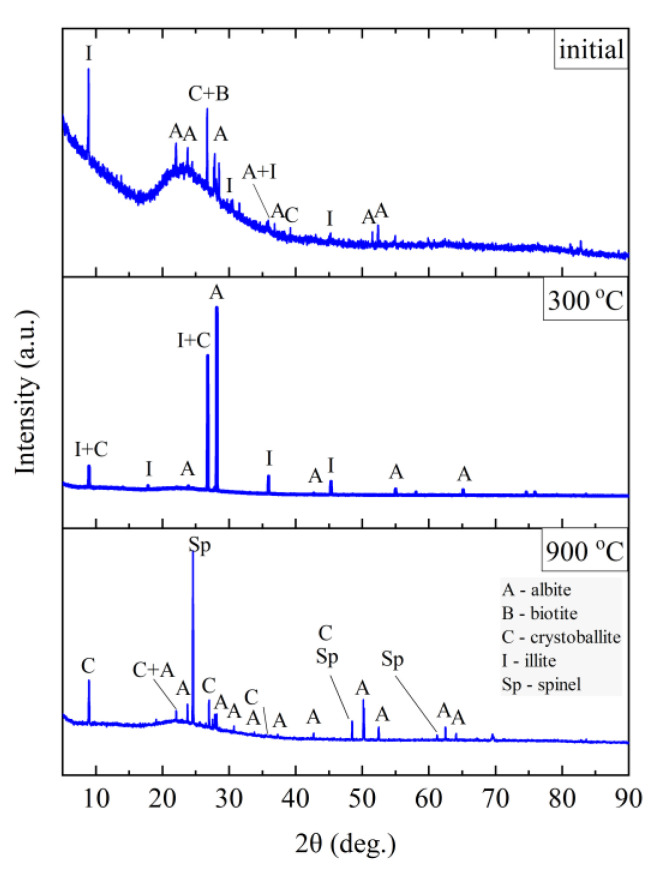
XRD diffractograms for Slovak perlite ore SP3.

**Figure 14 materials-14-02946-f014:**
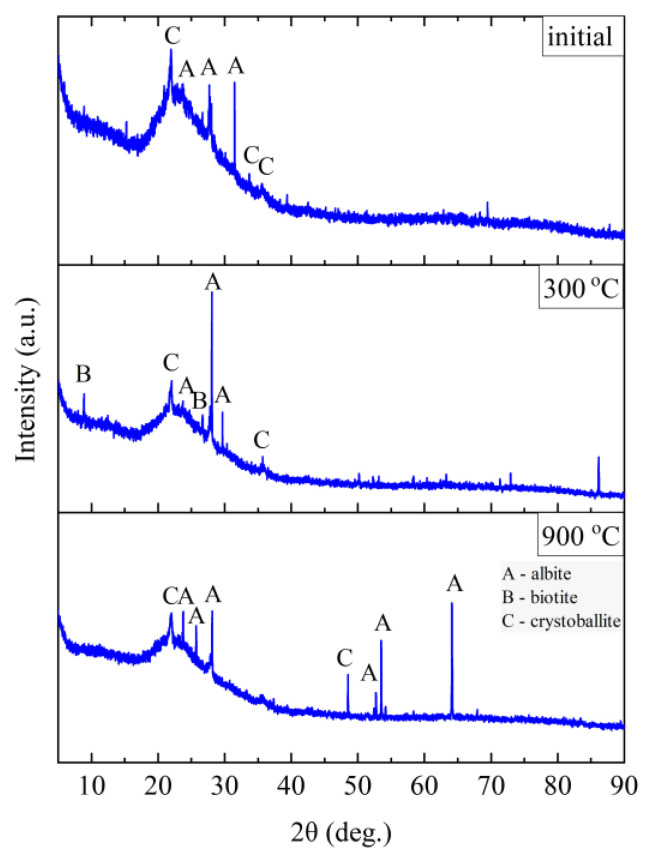
XRD diffractograms for Hungarian perlite ore HP.

**Figure 15 materials-14-02946-f015:**
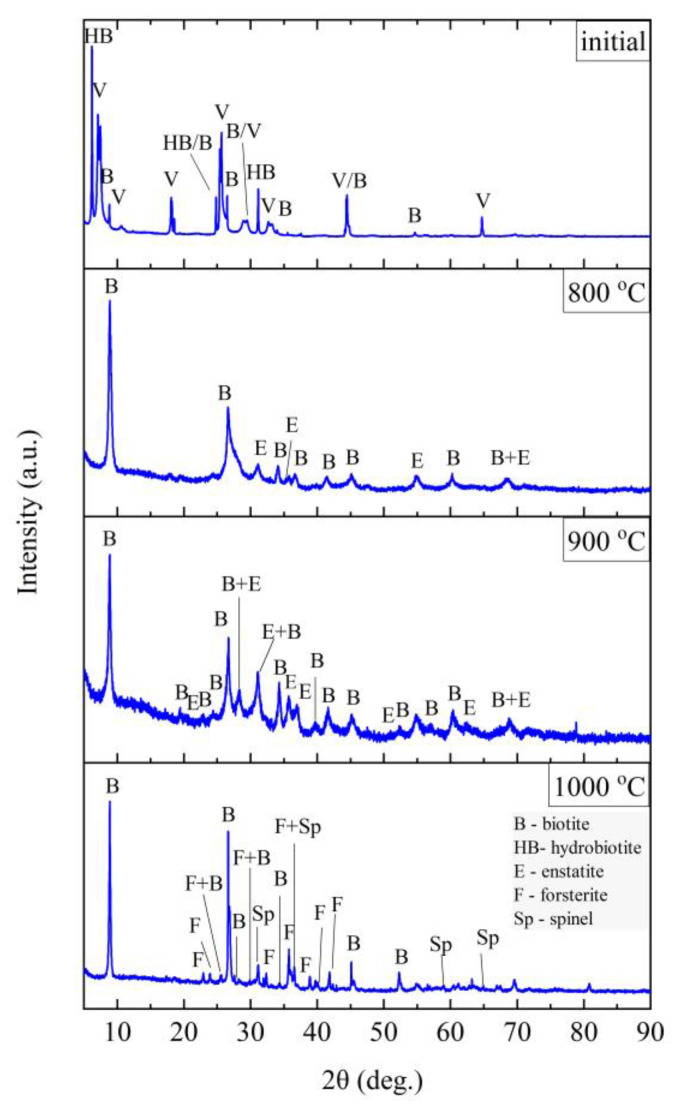
XRD diffractograms for vermiculite (V).

**Figure 16 materials-14-02946-f016:**
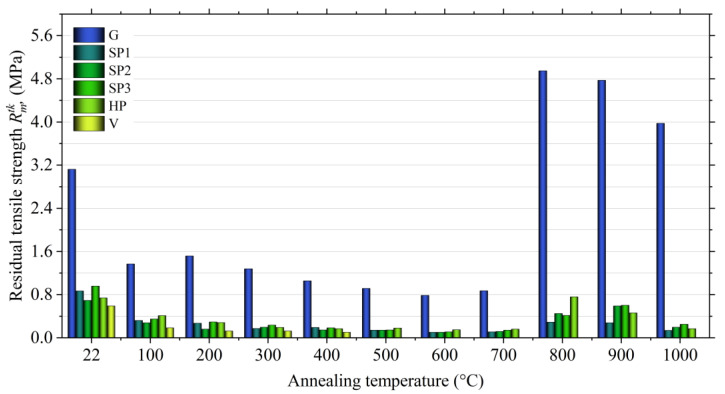
Final tensile strength ($R_{m}^{tk}$) of moulding sands with Geopol^®^ binder, depending on the type of additive.

**Figure 17 materials-14-02946-f017:**
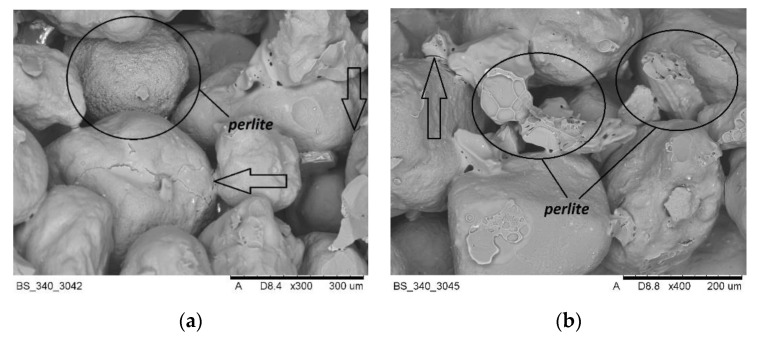
SEM imaging of moulding sand with SP1 perlite ore after annealing at 800 °C: (**a**) magnification 300×, (**b**) magnification 400×.

**Figure 18 materials-14-02946-f018:**
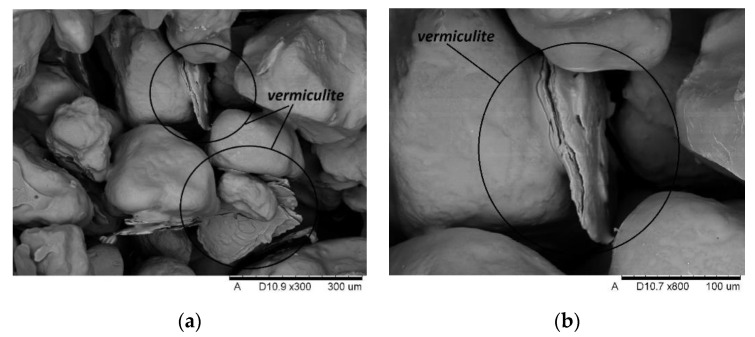
SEM imaging of moulding sand with vermiculite (V) after annealing at 800 °C: (**a**) magnification 300×; (**b**) magnification 800×.

**Table 1 materials-14-02946-t001:** Grain size analysis of Slovak perlite ore (SP1–SP3), Hungarian perlite ore (HP), ground vermiculite (V) and quartz sand (QS—matrix of moulding sands).

Mesh Size Sieve	Material					
	SP1	SP2	SP3	HP	V	QS
1.600	0.00	0.00	0.00	0.00	0.00	0.00
0.800	0.00	0.00	**0.95**	0.00	0.00	0.00
0.630	0.00	0.00	**1.58**	1.84	0.00	0.48
0.400	0.00	**48.16**	**29.98**	**30.08**	0.06	7.19
0.320	0.00	**30.21**	40.10	**20.62**	0.13	**18.63**
0.200	0.00	**11.58**	25.94	**29.09**	3.36	**50.21**
0.160	6.52	4.38	0.31	7.93	9.83	**13.10**
0.100	9.29	4.10	0.38	7.36	22.54	9.71
0.071	**9.35**	0.67	0.20	1.22	**13.34**	0.56
0.056	**22.03**	0.33	0.20	0.53	**3.66**	0.07
Bottom	**52.81**	0.57	0.36	1.33	**47.08**	0.05
Sum	100.00	100.00	100.00	100.00	100.00	100.00

**Table 2 materials-14-02946-t002:** Comparison of grain size distribution of tested materials.

Parameter	Unit	Material					
		SP1	SP2	SP3	HP	V	QS
Average grain size, *d_L_*	mm	0.08	0.30	0.31	0.24	0.06	0.23
Average grain size, *D*_50_	mm	0.10	0.40	0.36	0.33	0.07	0.26
The mesh size sieve where the main faction has gathered	-	0.071/ 0.056/ bottom	0.40/ 0.32/ 0.20	0.40/ 0.32/ 0.20	0.40/ 0.32/ 0.20	0.071/ 0.056/ bottom	0.32/ 0.20/ 0.16
Main fraction, *F_g_*	%	92.61	89.95	96.02	79.79	82.96	81.94
Distribution factor, *S*_0_	-	1.27	1.20	1.19	1.38	1.93	1.27
Inclination indicator, *S_k_*	-	1.04	1.04	1.05	0.93	0.98	1.00
Homogeneity degree, *GG*	%	61.00	68.00	74.00	50.00	21.00	68.00
Grain number, *L*	-	150.52	42.90	40.76	52.27	21.86	55.27

**Table 3 materials-14-02946-t003:** Elemental composition of the tested materials.

Component	SP1	SP2	SP3	HP	V
O	47.6	47.35	48.63	48.31	39.60
Si	33.18	33.18	33.88	34.58	15.96
Al	6.45	6.14	7.69	6.32	3.47
K	5.89	6.66	3.78	5.13	7.04
Fe	2.72	2.65	1.43	2.05	20.90
Ca	1.73	1.68	0.95	1.35	1.25
Na	1.26	1.36	2.81	1.61	0.11
Ti	0.32	0.21	0.12	0.11	1.89
Mg	0.30	0.16	0.37	0.08	8.61
Others	1.09	0.61	0.34	0.46	1.17

**Table 4 materials-14-02946-t004:** Oxide composition of the tested materials.

Component	SP1	SP2	SP3	HP	V
Na2O	1.70	1.84	3.79	2.18	0.14
MgO	0.49	0.27	0.61	0.14	14.27
Al2O3	12.20	11.60	14.53	11.94	6.55
SiO2	70.99	70.99	72.47	73.99	34.13
K2O	7.09	8.03	4.55	6.18	8.48
CaO	2.43	2.35	1.33	1.89	1.75
Fe2O3	3.89	3.80	2.04	2.93	29.88
TiO2	0.53	0.50	0.21	0.18	3.16
Others	0.68	0.62	0.47	0.57	1.64

## Data Availability

The data are contained within the article and/or are available on request from the corresponding author.

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
