# Peer review of "Dehydroxylation of Perlite and Vermiculite: Impact on Improving the Knock-Out Properties of Moulding and Core Sand with an Inorganic Binder"

_materials, 2021, doi:10.3390/ma14112946_

Round 1

Reviewer 1 Report

The manuscript is relatively well written and appears clear in content. The main limitation is the lack of significant innovation and the use of characterization techniques with limited scientific value. For example, SEM microscopy is used, but with a very low magnification that does not justify its use. In my opinion, as thermal treatment have been performed the deepening with some really structural investigations (FTIR is not), such as the X-rays diffraction with a qualitative and a quantitative approaches, should be used to reach a more significant level of innovation.

Author Response

Response to Reviewer 1 Comments

  1. “ The main limitation is the lack of significant innovation and the use of characterization techniques with limited scientific value”.

Response 1: Thank you for your suggestion. The topic was not properly motivated. The topic of the work was taken up after the literature review. It is difficult to find data on molding sands with an inorganic binder and satisfactory knock-out properties. It is the knock-out and low susceptibility to reclamation, that are the main limitations in their common application. The aim of the article was therefore, among other things, to draw attention to the fact, that it is possible to use additives reducing the final strength of the molding sands, while maintaining their basic advantage - low harmfulness to the environment. In our opinion, Slovak perlite is the least studied deposit in Europe. The available research results required considerable deepening. There were also no studies on the influence of fraction size on the chemical and phase composition and their relationship with thermal and structural studies. Literature data mainly concern Greek perlite, which, according to the authors, is the best-studied deposit in Europe. Due to the fact that the best effects of reducing the final strength of molding sands with a geopolymeric binder were obtained for vermiculite, detailed tests were also carried out for this material. There are no studies in the literature on molding sand with a geopolymeric binder and proposed mineral additives. The research methodology was supplemented with the analysis of the chemical composition (XRF) and the mineralogical composition (XRD).

  1. “For example, SEM microscopy is used, but with a very low magnification that does not justify its use”.

Response 2: thank you - perhaps its use has not been well justified. The aim of the research was not to show how perlite ore and vermiculite change under the influence of temperature, which is generally known, but to present the effect of their action in molding sands - the formation of microcracks. At higher magnification, we were not able to visualize this mechanism.

  1. “In my opinion, as thermal treatment have been performed the deepening with some really structural investigations (FTIR is not), such as the X-rays diffraction with a qualitative and a quantitative approaches, should be used to reach a more significant level of innovation”.

Response 3: Thank you for the suggestion as it turned out to be very helpful and significantly improved the scientific quality of the article. I agree with the statement that the FTIR method did not provide a complete picture of the structural changes. That is why we extended the scope of our research. First, the analysis of the chemical composition of the samples (XRF) was performed in order to correctly interpret the results of the XRD tests. Combining the results of thermal analysis, FTIR, XRD and XRF for perlite from various deposits, with different grain sizes and vermiculite, allowed to present interesting conclusions.

Additionally:

  • Improved theoretical introduction. It was enriched with a literature review related to the subject matter.
  • We gave up on microscopic photos taken with an optical microscope because they provided little information.
  • The description of FTIR results has been improved and linked to the results of thermal analysis and XRD.
  • The order of subchapters was changed so that the obtained results would determine the commencement of further research.
  • Linguistic revision was made and conclusions were modified.

Reviewer 2 Report

Dear Authors,

Congratulations on your work, which is interesting, but needs important improvements, such as:

  1. No contextualization of the research is properly made;
  2. The Introduction presents large groups of references, but no direct reference to other similar works is made. Moreover, no gaps in previous research have been identified, which is a severe lack in your work. Please spend time to describe in detail the main motivations of your work;
  3. No discussion is presented, comparing your results to others previously published;
  4. The conclusions don't refer how one ore more problems have been overcome due to your work;
  5. Please state the main novelty brought by your work;
  6. Please make a deep proof-read of your work, because there are systematic errors and sentences ill-constructed. Example: "numer" is not number.
  7. Please provide information about the software used in microscopy.

Good luck.

kind regards,

Reviewer

Author Response

Response to Reviewer 2 Comments

Point 1. “No contextualization of the research is properly made”

Response 1: Thank you for your suggestion. Agree - there was no reference to other works. An in-depth review of the literature was made and the current state of knowledge was described.

Point 2. “The Introduction presents large groups of references, but no direct reference to other similar works is made. Moreover, no gaps in previous research have been identified, which is a severe lack in your work. Please spend time to describe in detail the main motivations of your work;

Response 2: The motivation for undertaking the topic was described, which was an in-depth analysis of perlite ore from the Slovak deposit of various grain sizes and comparing it with the Hungarian deposit and vermiculite. The results of TG/DTG/DTA studies were related to FTIR, XRF and XRD. Such research has not been found in the literature.

Point 3. “No discussion is presented, comparing your results to others previously published;”.  

Response 3: Thank you - I agree with the opinion. I made the appropriate connections with previously published research results.

Point 4. “The conclusions don't refer how one ore more problems have been overcome due to your work”

Response 4: The requests have been extended and modified.

Point 5. “Please state the main novelty brought by your work”.

Response 5: It was indicated in the introduction that studies were conducted aimed at reducing the final strength of molding sands with inorganic binders, but they concerned, for example, modification of the structure of water glass or the introduction of expanded perlite into the sands with water glass. There is insufficient information in the literature on the impact of mineral additives such as raw perlite ore and vermiculite introduced into molding or core sand at the stage of their preparation, and the mechanism and measurable effect of their action - the use of dehydroxylation in contact with high temperature of the liquid metal and its accompanying rapid change volume. There is no information on the research of molding and core sand with geopolymer binder with the proposed additives. The work fills this gap. The work broadens the knowledge of the Slovak and Hungarian perlite deposits research. So far, there are no test results that would combine the TG/DTG/DTA analysis with the FTIR, XRD and XRF results, which were additionally performed. The size of the Slovak perlite fraction was also influenced by the change of its structure under the influence of heating. In addition, structural changes in South African vermiculite under the influence of temperature were identified and linked.

Point 6. “Please make a deep proof-read of your work, because there are systematic errors and sentences ill-constructed. Example: "numer" is not number”

Response 6: Thank you - the necessary correction has been made.

Point 7. “Please provide information about the software used in microscopy”.

Response 7: Thank you - a supplemented description of the research methodology has been introduced

Additionally:

  • We gave up on microscopic photos taken with an optical microscope because they provided little information.
  • Linguistic revision was made and conclusions were modified.

Reviewer 3 Report

Comments

  1. It is not clear what is the originality of this investigation
  2. The authors should present a better characterization of their starting materials (perlite and Vermiculite). I believe that XRD patterns of the starting materials are necessary in order to know if the samples are pure and if not what other minerals coexists and in what amounts. This is crucial in order to discuss the results properly.
  3. I believe that the authors can not present and discuss DTA/TGA results correctly if they do not have data about the quantitative mineralogical analysis of the samples. In their manuscript they discuss their data like the samples were pure, but are they? In the materials part they refer to impurities in a not clear way.
  4. The starting material is not described clearly in the case of perlite the impurities are referred generally to the deposits and in the case of vermiculite in general about vermiculite deposits.
  5. Vermiculite is not a smectite as the authors claim.
  6. SEM images and generally SEM study is not well presented. The SEM images are of low magnification. I believe that with these low magnification images the authors are not show what there are claiming especially for vermiculite.

Author Response

Response to Reviewer 3 Comments

Point 1. “It is not clear what is the originality of this investigation”

Response 1: Thank you for your suggestion. It was indicated in the introduction that studies were conducted aimed at reducing the final strength of molding sands with inorganic binders, but they concerned, for example, modification of the structure of water glass or the introduction of expanded perlite into the sands with water glass. There is insufficient information in the literature on the impact of mineral additives such as raw perlite ore and vermiculite introduced into molding or core sand at the stage of their preparation, and the mechanism and measurable effect of their action - the use of dehydroxylation in contact with high temperature of the liquid metal and its accompanying rapid change volume. The work fills this gap. The work broadens the knowledge of the Slovak and Hungarian perlite deposits research. So far, there are no test results that would combine the TG / DTG / DTA analysis with the FTIR, XRD and XRF results, which were additionally performed. The size of the Slovak perlite fraction was also influenced by the change of its structure under the influence of heating. In addition, structural changes in South African vermiculite under the influence of temperature were identified and linked.

Point 2. “The authors should present a better characterization of their starting materials (perlite and Vermiculite). I believe that XRD patterns of the starting materials are necessary in order to know if the samples are pure and if not what other minerals coexists and in what amounts. This is crucial in order to discuss the results properly”

Response 2: Thank you for the suggestion as it turned out to be very helpful and significantly improved the scientific quality of the article. That is why we extended the scope of our research. First, the chemical composition of the samples (XRF) was analyzed for the initial samples in order to correctly interpret the results of the XRD tests.  Combining the results of thermal analysis, FTIR, XRD and XRF for perlite from various deposits, with different grain sizes and vermiculite, allowed to present interesting conclusions.

Point 3. “I believe that the authors can not present and discuss DTA/TGA results correctly if they do not have data about the quantitative mineralogical analysis of the samples. In their manuscript they discuss their data like the samples were pure, but are they? In the materials part they refer to impurities in a not clear way;”.  

Response 3: Thanks for the suggestion. Of course I agree. Chemical composition (XRF) and phase (XRD) analysis was performed and related to the results of thermal and FTIR analyzes.

Point 4. “The starting material is not described clearly in the case of perlite the impurities are referred generally to the deposits and in the case of vermiculite in general about vermiculite deposits.”

Response 4: thank you - I reviewed the literature and presented better characterization of the materials.

Point 5. “Vermiculite is not a smectite as the authors claim.”.

Response 5: Of course I agree. The description was incorrect. The error has been corrected.

Point 6. “SEM images and generally SEM study is not well presented. The SEM images are of low magnification. I believe that with these low magnification images the authors are not show what there are claiming especially for vermiculite.”

Response 6: Thank you - perhaps its use has not been well justified. The aim of the research was not to show how perlite ore and vermiculite change under the influence of temperature, which is generally known, but to present the effect of their action in molding sands - the formation of microcracks. At higher magnification, we were not able to visualize this mechanism. A supplemented description of the SEM research methodology has been introduced.

Additionally:

  • We gave up on microscopic photos taken with an optical microscope because they provided little information.
  • Linguistic revision was made and conclusions were modified.

Reviewer 4 Report

The proposed study is well structured with very good analysis of the results. Minor revision is proposed. See below my comments:

Line 138: I suggest to express the binder composition in % percent to be more comprehensive by the reader. 

Line 151: The Hungarian perlite ore should be mentioned in this sentence since it is also examined for its tensile resistance.

Line 163: In Table 1 you have to correct the heading in the first column since you do not present sieves numbers but sieve openings (mm).

Line 229: I suggest to include the interpretation of the IR bands in the range of 1200-800 cm-1 as well as that of the new bands formed after the thermal treatment above 800oC, in order to present a more complete analysis of the structural transformations happening in the examined system.   

Line 241: correct pearlite to perlite

Line 242: correct figure to Figure 14.

Line 243: In this sentence, you have to add the Hungarian perlite ore.

Line 244: Mention in the materials section what binder is this.

Line 258: correct pearlite to perlite

Line 290: Unify the captions of figures 16 and 17. The micrographs show the same material in a different magnification. Do the same for figures 18 and 19.

Author Response

Response to Reviewer 4 Comments

Line 138: „I suggest to express the binder composition in % percent to be more comprehensive by the reader”. 

Response: Thank you - as suggested, the % share of molding sand ingredients has been introduced.

Line 151: The Hungarian perlite ore should be mentioned in this sentence since it is also examined for its tensile resistance.

Response: Of course, that statement was missing. The necessary correction has been made.

Line 163: In Table 1 you have to correct the heading in the first column since you do not present sieves numbers but sieve openings (mm).

Response: For the sake of clarity of the message, adjustments were made in Tables 1 and 2.

Line 229: I suggest to include the interpretation of the IR bands in the range of 1200-800 cm-1 as well as that of the new bands formed after the thermal treatment above 800oC, in order to present a more complete analysis of the structural transformations happening in the examined system.   

Response: Thank you - I agree with the suggestion. For full analysis, chemical composition (XRF) and phase (XRD) tests were performed and related to the TG/DTG/DTA and FTIR results. Changes occurring in the vermiculite sample above 800oC and in the wavenumber range of 1200-800 cm-1 were interpreted.

Line 241: correct pearlite to perlite

Response: The mistake has been corrected

Line 242: correct figure to Figure 14.

Response: the mistake has been corrected

Line 243: In this sentence, you have to add the Hungarian perlite ore.

Response: a correction was made.

Line 244: Mention in the materials section what binder is this.

Response: a correction was made.

Line 258: correct pearlite to perlite

Response: the mistake has been corrected

Line 290: Unify the captions of figures 16 and 17. The micrographs show the same material in a different magnification. Do the same for figures 18 and 19.

Response: a correction was made.

Additionally:

  • We gave up on microscopic photos taken with an optical microscope because they provided little information.
  • The scope of research was extended and the analysis of the results was deepened
  • The obtained results were related to the results of works by other authors
  • The layout of the work has been modified to be more logical and structured
  • Linguistic revision was made and conclusionswere modified.

Round 2

Reviewer 1 Report

I see the work has been extensively revised by the authors who removed the optical microscopy section and inserted XRF and XRD analyses. I believe that only some of the contents have been consistently implemented as required. In addition, some of the data are absent, others are redundant and unnecessary and do not help in a critical comparison of the data presented. I consider that the manuscript can be considered for publication in Materials after responding and implementing the manuscript as described below.

1). XRF analysis represents an important implementation for the evaluation of chemical composition. However, I think that the use of two digits after the decimal point in XRF measurements (Table 3 and 4) is excessive. In my opinion, it is more realistic to consider using one digit after the comma.

2). Many of the IR measurements seem unnecessary to me; the IR spectra shown in Figure 6 and Figures 7, 8, 9, and 10 are very similar, sometimes are even identical. For example, the only substantial difference of Slovak perlite ore (SP1) and of vermiculite vs. the other samples is that they only changed substantially between 900 and 1000°C heat treatments. I suggest removing figures 6, 7, 8, 9, 10 to be included in the supplementary material (supplementary data file).  I suggest adding a single figure for comparing the IR spectra collected at 25°C and after 1000°C of thermal treatment for the different samples, instead of Figure 6, 7, 8, 9, 10.

3). The XRD part is an important integration compared to the previous revision and can easily justify and add relevant content to the discussion in the commentary of the composition with the structure properties. For example, it can explain why there were variations in the IR spectra at high temperatures. In this regard, the missing spectra after treatment at 1000°C should be added to comment on the different vibrational characteristics.

4). I do not understand the reason why the authors do not introduce higher resolution SEM images according to the characteristics of the instrument and the reported magnification performances. I expect also from the optical images present in the before manuscript an abrupt change of morphology. It would be desirable to havea section containing a full comparison of the morphological properties at 25°C and after 1000°C of treatment for all the different samples.

5). Optical image section of the previous manuscript should be inserted in the supplementary data section.

6). please verify and standardize throughout the text the use of upper and lower case characters in the samples.

Author Response

Comments:

1) „XRF analysis represents an important implementation for the evaluation of chemical composition. However, I think that the use of two digits after the decimal point in XRF measurements (Table 3 and 4) is excessive. In my opinion, it is more realistic to consider using one digit after the comma”.

Response 1: Throughout the article, we apply the same accuracy in presenting the results. We propose to remain in such a convention in order to maintain consistency of content - eg. Table 1 and 2.

2) „Many of the IR measurements seem unnecessary to me; the IR spectra shown in Figure 6 and Figures 7, 8, 9, and 10 are very similar, sometimes are even identical. For example, the only substantial difference of Slovak perlite ore (SP1) and of vermiculite vs. the other samples is that they only changed substantially between 900 and 1000°C heat treatments. I suggest removing figures 6, 7, 8, 9, 10 to be included in the supplementary material (supplementary data file).  I suggest adding a single figure for comparing the IR spectra collected at 25°C and after 1000°C of thermal treatment for the different samples, instead of Figure 6, 7, 8, 9, 10”.

Response 2: We cannot agree with that. One of the elements of the publication is to show the effect of particle size distribution on the dehydroxylation process. The spectra serve this purpose and, according to the authors, show changes taking place under the influence of temperature at characteristic points, i.e. the wavenumber approx. 1630-1640 cm-1 and in the range of approx. 3650-3400 cm-1. Apart from that, apart from Slovak perlite, we also have Hungarian perlite. We believe that the results presented in this way are more reliable.

3) „The XRD part is an important integration compared to the previous revision and can easily justify and add relevant content to the discussion in the commentary of the composition with the structure properties. For example, it can explain why there were variations in the IR spectra at high temperatures. In this regard, the missing spectra after treatment at 1000°C should be added to comment on the different vibrational characteristics”.

Response 3: In our opinion, the research results are consistent and properly explained. The results of thermal tests were linked (as the Reviewer suggested in the first review) with the FTIR and XRD results. It was also indicated that the changes in the samples may result from the chemical composition, in particular the K2O / Na2O ratio (chapter 3.3). The IR spectra of samples annealed at 1000°C showed structural changes in the SP1 pearlite ore, therefore XRD results are presented for consistency. In the remaining samples, such changes were not registered as described: “ For the sample of Slovak perlite with the smallest grain size (SP1), the primary structure of the material decomposed at 1000°C (Fig. 11). The remaining samples, both Slovak and Hungarian, retain their structure, but a change in the half-width of the bands can be indicated as the temperature increases”.

Changes in the vermiculite sample were also shown. The results of the structural studies were linked to the XRD results and confirmed with the literature data: „According to literature data [51] at 900°C the illite decomposes and enstatite appears, and when heated to 1000°C, enstatite decomposes and forsterite appears. Such a course is indicated by the appearance of an intense band at 878 cm-1, characteristic of forsterite”.

4) I do not understand the reason why the authors do not introduce higher resolution SEM images according to the characteristics of the instrument and the reported magnification performances. I expect also from the optical images present in the before manuscript an abrupt change of morphology. It would be desirable to havea section containing a full comparison of the morphological properties at 25°C and after 1000°C of treatment for all the different samples.

Response 4: As already mentioned in response to the first review, SEM tests were performed to identify microcracks in the bridges connecting the grains of the grain matrix. Thus, they explain a clear decrease in the strength of the moulding sand with the proposed additives, subjected to a thermal load above the temperature of 800°C against the background of the moulding sand without additives (G) - Fig. 16. As already mentioned, the use of a higher magnification is pointless as it will not capture the principle of operation of crude perlite ore and crude vermiculite - before dehydroxylation. The study of the surface morphology in relation to the application described in the article seems to be pointless.

5) Optical image section of the previous manuscript should be inserted in the supplementary data section.

Response 5: Based on discussions with the co-authors of the article, we decided not to use microscopic photos taken with an optical microscope because they provided little information.

6) Please verify and standardize throughout the text the use of upper and lower case characters in the samples.

Response 6: Thank you. Necessary corrections were made.

Summary:

In line with the suggestions from the first review, the suggested research was carried out, which significantly increased the level of innovation in the article. FTIR, XRD and thermal analysis for various materials were related. The Slovak perlite ore was characterized in detail and the influence of the grain size on its behavior under the influence of high temperature was demonstrated.

Reviewer 2 Report

Congratulations for the upgrade done!

Now, a deep proof-read needs to be done.

Kind regards,

Reviewer

Author Response

Comment: „Now, a deep proof-read needs to be done”.

Response: Thank you for your positive review. We have made a thorough editorial correction.

Reviewer 3 Report

The authors did improve their manuscript significantly.

I would like to suggest that the XRD patterns especially in the case of clay minerals should be provided at least from 3o 2Theta. The authors provided from 5o 2Theta and therefore vermiculite 001 peak is not clearly observed. Additionally, there is no point on provide XRD patterns over 70o 2Theta.

Author Response

Comment: „I would like to suggest that the XRD patterns especially in the case of clay minerals should be provided at least from 3° 2Theta. The authors provided from 5° 2Theta and therefore vermiculite 001 peak is not clearly observed. Additionally, there is no point on provide XRD patterns over 70° 2Theta.”

Response: Thank you for your positive review. I agree with the Reviewer's suggestion. The diffractograms of samples should be the measurement from 3° of 2Theta. Due to the short deadline time of our responses for Reviewer, unfortunately, we are not able to repeat the measurement of sample diffractograms. In relation to the data presented  in the manuscript (XRD part), we concluded, that the samples have consisted of presented clay minerals. The obtained diffractograms allowed to capture characteristic reflections and identify crystalline phases. Reflections above 70° 2Theta were removed.

We have made a thorough editorial correction.